# Possible Interactions between Malaria, Helminthiases and the Gut Microbiota: A Short Review

**DOI:** 10.3390/microorganisms10040721

**Published:** 2022-03-27

**Authors:** Jean d’Amour Mutoni, Jean-Paul Coutelier, Nadine Rujeni, Leon Mutesa, Patrice D. Cani

**Affiliations:** 1Biomedical Laboratory Sciences Department, College of Medicine and Health Sciences, University of Rwanda, Kigali P.O. Box 3286, Rwanda; mutoni.jeandamour@gmail.com (J.d.M.); jean-paul.coutelier@uclouvain.be (J.-P.C.); nrujeni@gmail.com (N.R.); 2De Duve Institute, Université Catholique de Louvain (UCLouvain), 1200 Brussels, Belgium; 3Centre for Human Genetics, College of Medicine and Health Sciences, University of Rwanda, Kigali P.O. Box 3286, Rwanda; lmutesa@gmail.com; 4Metabolism and Nutrition Research Group, WELBIO—Walloon Excellence in Life Sciences and BIOtechnology, Louvain Drug Research Institute, Université Catholique de Louvain (UCLouvain), 1200 Brussels, Belgium

**Keywords:** malaria, *Plasmodium*, helminthiases, STHs, gut microbiota

## Abstract

Malaria, caused by the *Plasmodium* species, is an infectious disease responsible for more than 600 thousand deaths and more than 200 million morbidity cases annually. With above 90% of those deaths and cases, sub-Saharan Africa is affected disproportionately. Malaria clinical manifestations range from asymptomatic to simple, mild, and severe disease. External factors such as the gut microbiota and helminthiases have been shown to affect malaria clinical manifestations. However, little is known about whether the gut microbiota has the potential to influence malaria clinical manifestations in humans. Similarly, many previous studies have shown divergent results on the effects of helminths on malaria clinical manifestations. To date, a few studies, mainly murine, have shown the gut microbiota’s capacity to modulate malaria’s prospective risk of infection, transmission, and severity. This short review seeks to summarize recent literature about possible interactions between malaria, helminthiases, and the gut microbiota. The knowledge from this exercise will inform innovation possibilities for future tools, technologies, approaches, and policies around the prevention and management of malaria in endemic countries.

## 1. Introduction

With 241 million cases and 627,000 deaths reported in 2020, malaria remains a huge global morbidity and mortality issue [1]. Helminths, affecting one-third of the population in developing regions of the world, pose more morbidity than mortality problems [2]. Importantly, the two are often co-morbid. Global efforts to prevent and manage these diseases continue to evolve with new innovations. Harnessing the gut microbiota is one of the newest and most promising approaches. Research evidence has shown its important link with health and disease [3,4]. This review explores known facts and gaps in knowledge about interactions between malaria, helminthiases, and human gut microbiota. 

## 2. Malaria

Malaria is a parasitic disease that, for many years, has remained a major cause of morbidity and mortality in the developing world. According to the World Health Organization (WHO), reported malaria morbidity cases in the previous decade show concerning trends: 251 million in 2010, 216 million in 2016 [5], 231 million in 2017, 228 million in 2018 [6], and 241 million in 2020 [1]. Similarly, recorded deaths were 585 000 in 2010, 416,000 in 2017, 405,000 in 2018 [6], and 627,000 in 2020 [1]. Consistently representing more than 90% of cases and deaths, sub-Saharan Africa is the most affected region [5], and young children and pregnant women are the most vulnerable groups [7]. Therefore, through its global technical strategy for malaria 2016–2030, the WHO presents malaria as a global health priority [8]. 

Malaria is caused by an apicomplexan parasite of the genus Plasmodium, which is transported by a female Anopheles mosquito from human to human. Plasmodium falciparum, the most common causative species in endemic areas, is believed to cause most deaths [9]. 

There are three clinical presentations of malaria: severe or complicated, mild or uncomplicated [10], and asymptomatic [9]. The human response to malaria varies depending on factors such as the genetics of parasite proteins, co-infections, comorbidities, host nutrition, host genetics, host ethnic background, geographical locations, and microbiota composition, to name a few [11,12]. That response starts with physical barriers (e.g., skin), progresses to an innate immune response such as dendritic cells, and culminates in more adaptive responses (e.g., cytophilic antibodies) corresponding to the progress from infection to blood-stage disease [13]. 

As this short review is interested in malaria clinical outcomes, we will focus our discussion on the host immune response to blood-stage parasites. As reviewed by Crompton et al., that response involves (1) RBC sequestration and (2) systemic inflammation induced by the production of pro-inflammatory cytokines and chemokines (e.g., interleukin (IL)-1β, IL-6, IL-8, IL-12(p70), interferon (IFN)-γ, and tumor necrosis factor-α (TNF)-α), hemozoin, plasmodium proteins (e.g., plasmodium falciparum erythrocyte membrane protein 1 (PfEMP1) and merozoite surface protein-1 (MSP1), innate immune cells (e.g., natural killers (NKs), mast cells, neutrophils, etc.), and antibodies [14]. More specifically, a review by Waide et al. has shown that the lack of interleukin-21 (IL-21) signaling within the spleen leads to impaired clearance of *Plasmodium* infection [15]. Therefore, since the spleen germinal center (GC) is key to the activation and coordination of humoral immunity, it is also central to *Plasmodium* infection clearance. In the same direction, another review suggests that the clinical presentation of malaria, including the development of severe cases, appears to depend on a balance between pro-inflammatory and regulatory molecular and cellular mechanisms, although the precise role of regulatory cytokines (IL-10, transforming growth factor beta (TGFß)) and cells (regulatory T lymphocytes, IL-10-producing T lymphocytes) has not been completely solved [16]. 

On the other hand, genetic factors play a critical role. For example, considering ethnic background, a study conducted in Burkina Faso showed that two sedentary ethnic groups (Mossi and Rimaibé) were more vulnerable to mild malaria than the Fulani (nomad pastoralists) who, in fact, showed some form of resistance to severe malaria, evidenced by high parasitemia levels and a stronger antibody response [17]. 

It is also worth mentioning that correlations between several SNPs (Single Nucleotide Polymorphisms) in regulatory or coding regions of relevant genes and severe malaria were reported, despite the ongoing conflicting findings [9,18]. 

To date, there is only one WHO-recommended vaccine against malaria, RTS,S, which is also known by its trade name, ‘Mosquirix’ [19]. To accelerate progress, the WHO recommends the following multisectoral approaches to fight malaria: equitable health services, accurate surveillance and response, strategies tailored to local transmission settings, increased financing, political leadership, and universal health coverage [6]. 

## 3. Helminthiases

Helminths have co-existed and evolved with human intestinal bacteria since the dawn of time [20]. Research has shown the presence of helminth eggs in mummified human feces preserved for thousands of years [2]. The WHO technical report series on research priorities for helminth infections estimates that helminthiases are among the top causes of morbidity, mortality, and vicious poverty worldwide [21]. They are commonplace and affect at least one in three people in sub-Saharan Africa (SSA), Asia, and the Americas [2]. Helminthiases are diseases related to poverty, and they affect school children disproportionately compared to other age groups, hence negatively affecting their education performance [2].

Human helminth infections of public health importance are caused by species belonging to three major groups: nematodes, trematodes, and cestodes [2]. Nematodes, also known as round worms, are soil-transmitted (except for filarial ones) [2], whose main species of medical importance include: *Ascaris lumbricoides*, *Trichuris trichiura, Enterobius vermicularis, Toxocara canis, Necator americanus, Ancylostoma duodenale* (hookworm), *Strongyloides stercolaris, Wuchereria bancrofti (filarial), Loa loa (filarial), Dracunculus medinensis (filarial), Onchocerca volvulus (filarial), and Trichinella spiralis* (muscle tissue worm) [2,22]. Not all of them are endemic across the globe. For example, in SSA, common species include hookworms, *A. lumbricoides*, and *T. trichiura* [23]. Trematodes, also known as platyhelminth flukes [2], are flatworms and include the following species of medical importance: *Schistosoma mansoni, Schistosoma haematobium, Schistosoma japonicum, Fasciola hepatica, Clonorchis sinensis, Opisthorchis spp.,* and *Paragonimus spp* [2,22]. Finally, Cestodes, also known as platyhelminth tapeworms, include the following medically important species: *Taenia solium*, *Taenia sagina,* and *Echinococcus glanulosus* [2]. 

Helminth-malaria co-infections are very common since both diseases overlap geographically. A recent systematic review with meta-analysis reported a pooled prevalence of 17% of *plasmodium*-STH co-infection across Africa (central, east, west, and south parts), Southeast Asia, and South America [24]. However, the effects of helminth infections on malaria clinical manifestations remain uncertain [24,25,26]. For example, a study conducted in Ethiopia on 702 patients, with 19.4% infected with both *Plasmodium* and intestinal helminths, revealed that Soil-Transmitted Helminths (STH) infections were positively associated with non-severe malaria [27]. These findings are similar to those of a study conducted in Thailand where the risk of developing malaria was associated with the number of STH species infecting the participants [28]. On the other hand, a study conducted in Cameroon reported a malaria-helminth co-infection of 11.6% (*n* = 405) and no association between helminth infection and malaria clinical outcomes [29]. Finally, two other studies, both conducted in Ethiopia, showed neutral interactions, in terms of immune responses to malaria, between malaria and helminths [26,30].

## 4. The Gut Microbiota 

Thanks to the development of high throughput metagenomic sequencing, the last 15 years have been characterized by an incredible awareness of the potential of the gut microbiota. Although gut microbes in general have been studied for over 100 years, the investigation of the role of microorganisms that reside in the human gut has attracted much attention beyond classical infectious diseases and immunity [3,4]. Nowadays, the gut microbiota is investigated in many contexts including that of host metabolism, intestinal inflammatory bowel diseases, neurodegenerative diseases, and even cancers [31,32]. 

The gut microbiota is considered to be the collection of all taxa that constitute microbial communities, such as bacteria, archaea, fungi, and protists within the intestinal tract [3]. At the phylum level, it is commonly accepted that six main phyla dominate the microbiota: Firmicutes, Bacteroidetes, Actinobacteria, Proteobacteria, Fusobacteria, and Verrucomicrobia [33,34]. 

Besides the composition and the vast abundance of the microbial cells (10^14^) as compared to our own human cells (10^13^) [3], the metabolic activities of these gut bacteria are of utmost importance. Indeed, gut microbes are capable of producing various metabolites that are able to communicate with our cells [35]. It is worth noting that both the composition and the metabolites produced by the gut microbiota are strongly influenced by different endogenous and environmental factors such as diet, food additives and contaminants, genetics, immunological status, genetics, ethnicity, geographical location, gender, age, BMI, stool consistency, alcohol consumption, and drugs (e.g., antibiotics, proton pump inhibitors, metformin, and statins) [34,36,37,38,39]. 

Recent studies have also pointed out that the gut microbiota could shape the host immune response in different contexts, including the response to different immunomodulatory medications (e.g., immune checkpoint inhibitors, anti-proliferative drugs, and inflammatory cytokine inhibitors) [40]. Hence, the gut microbiota is now considered a key factor in contributing to the overall response of the host metabolism and immune response (Figure 1).

## 5. Interactions: Malaria, Helminthiases, and the Gut Microbiota

The gut microbiota is an important modulator of the natural immune system. In that sense, it plays a critical role in the control of pathogens and pathobionts, whereby the failure to do so leads to dysbiosis [41]. Moreover, if harnessed well, that community of gut commensals has been shown to be directly linked with better outcomes in the prevention, control, and management of different dysbiosis conditions: from cancer, to bloodstream infections, to obesity, to atopic infections, to mental health, and autoimmune arthritis [35,42,43,44,45]. 

Although the gut microbiota is largely investigated in the context of different bacterial infections, there are a limited number of publications about the impact of parasites on the composition of the gut microbiota. Still, some helminths’ interactions with the host’s gut microbiota have been documented [20]. For example, a study has shown that chronic *Trichuris muris* can decrease the host’s gut microbiota while specifically increasing the importance of lactobacilli in the intestinal bacterial community [46]. 

### 5.1. The Gut Microbiota Modulates Plasmodium Infection Risk and Transmission

The malaria life cycle in humans starts with the migration of its causative agent *plasmodium* from the female anopheles’ salivary gland into the mammalian skin. At that stage, transmitted ‘sporozoites’ have a glycan called ‘Gala1-3Galb1- 4GlcNAc-R (α-gal)’, which they share with a member of the gut microbiota community ‘*E. coli* O86: B7’. In their study, Yilmaz et al. demonstrated how *E. coli* O86: B7 elicits the production of anti–α-gal Abs, which block the transmission of *P. berghei* sporozoites from the mosquito to beyond the skin within mice (Figure 1) [47]. 

Applied to humans, the same antibodies showed a capacity to reduce the prospective risk of infection. A cohort of 195 Malian children and adults was enrolled in a study just prior to an intense *P. falciparum* transmission season to assess whether specific compositions of the human gut microbiota can modulate the risk of malaria infection. In their report, Yooseph et al. argued that the strategic modulation of commensals in the human intestinal tract could decrease the risk of *P. falciparum* infection in malaria-endemic areas [48]. According to the authors, a favorable composition of the gut microbiota includes three key members of the gut bacterial community: *Bifidobacterium*, *Streptococcus*, and the family Ruminococcaceae. Thus, they proposed that the administration of pre- and/or pro- biotic nutritional supplements could promote the growth of commensal organisms such as *Bifidobacterium*, which would make a difference in the context of malaria prevention (Table 1). 

### 5.2. The Gut Microbiota Modulates Malaria Disease Severity 

In a recent study conducted on children in Uganda, the authors demonstrated that gut microbiota, specifically bacteria, have the capacity to modulate the severity of malaria through regulating the spleen germinal center [49]. The same study was extended to testing the effects of antibiotics on gut microbial communities in ways that affect the host’s immunity to malaria. Surprisingly, dependent on pre-existing gut microbiota composition, antibiotics were shown to lead to changes in the gut microbiota composition that enhance resistance to *Plasmodium* infection [49]. Villarino et al. [50] demonstrated that specific members (*Lactobacillus* and *Bifidobacterium*) of the gut microbiota were responsible for the modulation of the severity of malaria in mice. Two groups of genetically similar mice from different commercial vendors were infected with the *Plasmodium* species. Mice from one vendor emerged as ‘resistant’, whereas others were ‘susceptible’ to infection. The analysis of cecal bacterial communities of the resistant group showed two distinguished population members: *Lactobacillus* and *Bifidobacterium*. When germ-free (GF) susceptible mice were treated with cecal content rich in *Lactobacillus* and *Bifidobacterium* or equivalent probiotics, the parasite burden dropped [50]. This study suggests that specific types of gut microbiota or treatment with supplements rich in *Lactobacillus* and *Bifidobacterium* have the potential to reduce the plasmodium parasite burden, which leads to the modulation of malaria severity (Figure 1) (Table 1). 

### 5.3. Malaria-Modulated Changes on Gut Microbiota

Malaria’s effects on the composition of the gut microbiota remain poorly understood and contradictory. On the one hand, the gut microbiota has been shown to decrease resistance to intestinal colonization of non-typhoidal *Salmonella* (NTS), and malarial immune effects may promote susceptibility to disseminated NTS infections [51]. Similarly, *P. berghei* ANKA (PbA) infection was demonstrated to cause intestinal and cerebral pathologies through modifying the gut microbiota whose Firmicutes decreased, whereas the proteobacteria increased [52]. Furthermore, in another study, mice infected with *P. berghei* ANKA showed a gut microbiota composition whose Bacteroidetes and Verrucomibrobia decreased, whereas Proteobacteria and Verrucomicrobia increased [53]. On the other hand, in a study that involved one hundred Kenyan infants, the authors ruled out the idea that *Plasmodium* infections and antimalarial treatment do not lead to major changes in human gut microbiota [54]. Surprisingly, the number of malaria episodes (two and beyond) caused significant variation in the gut microbiome composition (Table 1). 

### 5.4. Helminths and the Gut Microbiota

Interactions between intestinal helminths and intestinal bacteria in human hosts are inevitable. For instance, helminths can induce gut dysbiosis either directly via physical contact, through the production of chemical products and competition for nutrients, or indirectly through the modulation of host immune responses [55] (Figure 1). A systematic review and meta-analysis of studies from a varied geographical distribution reported that helminth infection significantly influences the alpha and beta diversity of the fecal microbiome [56]. In a study conducted on *Gasterosteus aculeatus* fish infected by *Schistocephalus solidus*—a natural parasitic helminth of fish—changes in the gut microbiota composition were shown to depend on parasite exposure [55]. 

Evidence from in vitro and murine studies shows that members of the gut microbiota (*Lactobacillus* spp. and *Enterococcus faecium*) contributed to a strengthening in the host immune responses against *Giardia* infection [57]. However, the gut microbiota, administered in the form of probiotics, was shown to play a role in modulating host-mediated inflammation and allergic responses to helminth infection rather than hampering or stopping the infection altogether [57]. 

A recent study conducted on non-human primates argued that changes in the gut microbiota composition (richness and diversity) were affected by the presence or absence of helminths. Nevertheless, bacterial richness increased and decreased with the presence of *Trichuris* sp. and *Strongyloides* sp., respectively [58], suggesting that the helminth species determines the direction of this relationship.

**Table 1 microorganisms-10-00721-t001:** Summary of different studies and their key findings about interactions between malaria and the gut microbiota.

Players (Host)	Key Findings	References
*P. yoelii*, gut microbiota and NTS (mice)	*P. yoelii* infection, through modifying the gut microbiota, decreases resistance to intestinal colonization of non-typhoidal salmonella (NTS) and malarial immune effects may promote susceptibility to disseminated NTS infections. Increase in Firmicutes and decrease in Bacteroidetes plus short-lived reduction in Proteobacteria were reported.	[51]
*P. berghei* and gut microbiota (mice)	Malaria infection was demonstrated to cause intestinal and cerebral pathologies through modifying the gut microbiota shown by the decrease in Firmicutes and increase in Proteobacteria.	[52]
*P. yoelii* and gut microbiota (mice)	Genetically similar mice purchased from two different vendors have shown different susceptibility to malaria due to their different gut microbiota composition.	[59]
*P. berghei ANKA infection and microbiota (mice)*	*P. berghei* ANKA infected mice show higher number of fecal *Acinetobacter*, *Lactobacillus*, and *Lachnospiraceae*_NK4A136_group.	[53]
*P. berghei, P. yoelii and* gut microbiota (mice)	The gut microbiota elicits the productions of anti–α-gal antibodies against malaria’s causative agent’s sporozoites which leads to blocking transmission.	[47]
*P. berghei, P. chabaudi, P. yoelii* and gut microbiota (mice)	Specific members (*Lactobacillus* and *Bifidobacterium*) of the gut microbiota were responsible for the modulation of the severity of malaria.	[50]
*P. falciparum* and gut microbiota (humans)	Specific compositions (*Bifidobacterium*, *Streptococcus*, and family Ruminococcaceae) of the human gut microbiota can modulate the risk if malaria infection.	[48]
*P. falciparum* and gut microbiota (humans)	Malaria episodes and Artemether-Lemefantrine (AL) treatment did not cause major changes in the gut microbiota composition.	[54]

## 6. Discussion 

Interactions between malaria and the gut microbiota have been studied more in mice than in humans (Table 1). Both players have been shown to affect each other directly and indirectly. For example, on one hand, in a murine study, malaria-induced modification of the gut microbiota composition led to the host’s increased vulnerability to non-typhoidal *Salmonella* infections [51]. On the other hand, differences in gut microbiota composition were shown to provide varied protection against malaria infection, including its severity modulation, in mice [59]. However, a recent study conducted in Kenyan infants infected with and treated for malaria concluded that human malaria episodes and Artemether-Lemefantrine (AL) treatment do not cause major changes in the gut microbiota composition [54]. By comparing the gut microbiota composition of samples taken before and after malaria infection / AL treatment, the authors observed no major changes in bacterial communities. The only reported changes, described by the authors as minimal, were observed by comparing samples taken from malaria episodes one and two [54].

The most significant study in this area has demonstrated the capacity of some members of the gut microbiota to elicit the production of anti-α-gal antibodies, which bind on *plasmodium* sporozoites to block the parasite’s transmission [47]. Similarly, another study conducted in humans demonstrated the capacity of specific gut microbiota compositions to modulate the prospective risk of malaria infection [48]. Furthermore, once the disease is established, a study conducted in mice has shown that specific members of the gut microbiota can modulate the severity of malaria [50]. 

The role of helminthiases in the tripartite (malaria, gut microbiota, and helminth) interactions is poorly studied, although it may be due to ecological reasons. Although malaria and helminthiases are both endemic and often co-morbid in the developing world, the human gut microbiota presents new possibilities to harness health outcomes. Thus, studying interactions among the three can generate findings that could inform decision-making in the fight against malaria and neglected diseases in endemic zones. 

Previous studies have shown the promise of the gut microbiota to: (1) reduce the prospective risk of malaria infection, (2) block its infection, and (3) modulate its severity. However, that promise still needs enough evidence to lead to tangible results. 

More powerful studies are necessary to determine the ability of the gut microbiota to modulate malaria outcomes in mammalian hosts. As discussed by Cani et al., microbiome-based large-scale studies are the gold standard and should be mandatory when one wants to characterize shifts in the gut microbial community composition [39]. 

Mandal et al. [54] suggest studies on the following topics: (1) gut microbiota differences between healthy individuals and patients with severe malaria, (2) interplay between gut microbiota and cerebral malaria, and (3) the potential of machine learning algorithms to predict the gut microbiota composition associated with children prone to *Plasmodium* infections. Yooseph et al. [48] suggest that future studies should explore the potential interactions between specific components of the microbiota, innate/adaptive immune responses, and susceptibility to malaria and other pathogens that commonly affect populations. The clinical outcome of malaria strongly depends on the type and intensity of immune responses, especially on the balance between pro-inflammatory and regulatory responses [16,60]. Since the gut microbiota is known to significantly modulate immune responses, including regulatory mechanisms [61,62,63] and antibody production [15], it may be hypothesized that modifications in the gut microbiota induced by environmental factors may, such as those shown in mice [50], affect malaria clinical presentation through immune mechanisms (Figure 1). 

In conclusion, despite limited published data, it has been shown that interactions between the gut microbiota, helminths, and malaria are indeed well established and may lead both to the modification of the gut microbiota, as well as to distinguished malaria outcomes. However, the available data remain largely insufficient for researchers to draw any conclusions, although they can chart a new path towards testing the hypothesized role of the gut microbiota in determining health and disease in malaria endemic settings. Therefore, more powerful and larger studies are recommended to help the scientific community make sense of this important topic. 

## Figures and Tables

**Figure 1 microorganisms-10-00721-f001:**
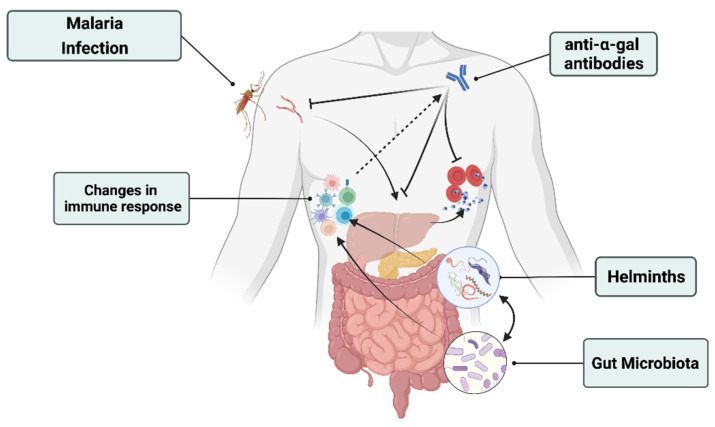
Helminths–gut microbiota–malaria interactions in the context of malaria transmission and severity. Malaria and helminths infections change the microbiota composition. Specific members of both the gut microbiota and helminths have been shown to modulate the severity of malaria in rodents by acting on immunity. For example, helminths and specific gut microbes contribute to stimulate the production of anti α-gal antibodies. These antibodies have been shown to stop the transmission and to decrease the risk of malaria transmission.

## Data Availability

We searched PubMed, Google Scholar and Global Health for publications related to the interactions between malaria, helminthiases, and gut microbiota. Our search used the following keyword combinations: ‘malaria + helminths + gut microbiota’, ‘malaria + helminths’, ‘malaria + gut microbiota’, ‘helminths + gut microbiota’, ‘plasmodium + helminths + gut microbiota’, ‘plasmodium + helminths’, ‘plasmodium + gut microbiota’, ‘malaria + STH + gut microbiota’, ‘malaria + STH’, ‘STH + gut microbiota’, ‘malaria + helminth + gut microbiome’, ‘malaria + gut microbiome’, ‘helminth + gut microbiome’, and ‘malaria + parasites + gut microbiota’. Through this search method, 21 relevant articles were identified and selected after removing duplicates.

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
