# Peer review of "Possible Interactions between Malaria, Helminthiases and the Gut Microbiota: A Short Review"

_microorganisms, 2022, doi:10.3390/microorganisms10040721_

Round 1

Reviewer 1 Report

please, specify in the title it is a review

please specify acronyms each time at first appearance

please specify what type of review it is

please report the methods section of this review

please, specify if the protocol review has been registered in advance, if not please clearly state it in the manuscript

please specify the search strategy

since malaria and helminthiases are still a relevant public health issues, please add considerations on the impact of your results. what implications?

Author Response

We would like to thank this reviewer for the comments. We have addressed all the questions.

  1. please, specify in the title it is a review: we added ‘A short review’ to the title.
  2. please specify acronyms each time at first appearance: all acronyms are now specified.
  3. please specify what type of review it is: we add ‘A short review’ to the title.
  4. please report the methods section of this review: a methods section has been added at the end of the text before references.
  5. please, specify if the protocol review has been registered in advance, if not please clearly state it in the manuscript: given that ours is a review and not a research study we did not register the protocol.
  6. please specify the search strategy: the search strategy is now specified under the methods section
  7. since malaria and helminthiases are still a relevant public health issues, please add considerations on the impact of your results. what implications? We did not generate any results as it is a short literature review, but we have discussed the link with public health and revised the numbers/incidence with the last updated data from 2020

Reviewer 2 Report

This is a review article that seeks to explore the interactions between helminth infection, the gut microbiota and malaria infection. Although there are interesting snippets of information in each of the ‘legs’ of this triangle – the information is not brought together in a comprehensive manner. The reader is left with the impression that there may or may not be interactions, which is not ideal.  Helminth contributions to changes in malaria infection or severity (either directly or through changes in the gut microbiota) need to be further discussed. There is at least one metareview on the topic (listed below). Without further exploration of this ‘leg’ of the triangle – it does not make sense to include the helminths in the review at all. They play a very minor role in the current manuscript.

The quality of the malaria pathogenesis section is questionable. There are only a very few topics touched upon – and most in the malaria field would agree that they are not the factors most important in pathogenesis. Recruitment of a co-author with a strong background in the malaria pathogenesis and subsequent reworking of this section would greatly improve the manuscript.

Several issues – major and minor are listed below in order of appearance (by Line number) in the manuscript.

This is listed in my copy as an “Article”, I suspect it should be listed as a “Review”.

14 – “nearly a million deaths” is overstated

29 - Plasmodium prevalence data is old. There are data from 2020 available.

49 – The authors should focus on focus on mortality and not infection when comparing the species of Plasmodium – as P. vivax has very high prevalence but low morbidity and mortality.

51 – Malaria severity is really a spectrum and not distinct entities. If, however, it needs to be classified into groups, “simple malaria” is not a term that is used in the field.

53 - Age and immune status are also factors that are important in determining infection outcome and should be included in the list.

55 – The authors seem to be mixing the innate immune response with physical barriers. They are quite distinct.

The malaria pathogenesis section is spotty; the input of a malaria expert as co-author is needed. SNPs and IL-21 are likely only minor players in pathogenesis and should not be mentioned in the absence of other factors (unless these are going to be tied into helminths or microbiota)

68-69 – “Environmental factors” seem to be conflated with genetics in this section.

88 – This statement about helminths “Being parasites” and therefore result in damage is backwards thinking. The characteristics of the organism is considered and then given a classification.

In the paragraph starting Line 111 discussing helminths and malaria. There is a nice meta-review in PLoS NTD by Afolabi that should be discussed.

173 – I do not see any reference for this statement. The referenced paper (43) does not look at the presence of this antibody and specifically states that the role of the anti-gal antibodies cannot be assessed in their study.

Lines 212 and following. This sentence is very awkwardly worded – I believe it is a double negative and therefore not stating what the authors wish.

Line 221 to 223 – Were these fish parasitized by a helminth? This sentence and its relevance to the discussion is unclear.

Table 2 – separate out by studies in mice first followed by all the studies in humans

Line 240 Studies have been conducted in vitro? I am not sure that you listed any of those. The majority of the studies seem to have been conducted in mice.

Line 244 “gut” typo

Lines 245 - 247 Again the wording is awkward and not likely as meant.

Line 248 – I am not sure that this was the ‘most significant finding’. I believe the authors should describe all of the findings and allow the reader to make their own decisions as to what is significant.

Author Response

First of all, we would like to thank this reviewer for the very useful comments that have indeed improved our review. We have addressed all the comments and suggestions. Our answers are listed below in bold.

This is a review article that seeks to explore the interactions between helminth infection, the gut microbiota and malaria infection. Although there are interesting snippets of information in each of the ‘legs’ of this triangle – the information is not brought together in a comprehensive manner. The reader is left with the impression that there may or may not be interactions, which is not ideal.  Helminth contributions to changes in malaria infection or severity (either directly or through changes in the gut microbiota) need to be further discussed. There is at least one metareview on the topic (listed below). Without further exploration of this ‘leg’ of the triangle – it does not make sense to include the helminths in the review at all. They play a very minor role in the current manuscript.

The quality of the malaria pathogenesis section is questionable. There are only a very few topics touched upon – and most in the malaria field would agree that they are not the factors most important in pathogenesis. Recruitment of a co-author with a strong background in the malaria pathogenesis and subsequent reworking of this section would greatly improve the manuscript.

>>> We have taken into account these remarks and revised the manuscript on each of the different sections. We also clarified the role of each of the factors (microbiota/helminths/malaria) and also highlighted whether the current knowledge was strong enough or needed additional studies. 

As suggested, we have now invited Prof. Léon Mutesa specialist of malaria to review, and contribute to the writing of our short review. 

Several issues – major and minor are listed below in order of appearance (by Line number) in the manuscript.

14 – “nearly a million deaths” is overstated: this was corrected and renamed: ‘…more than 600 thousand deaths…’

29 - Plasmodium prevalence data is old. There are data from 2020 available. This has been fixed through using data from 2020.

49 – The authors should focus on focus on mortality and not infection when comparing the species of Plasmodium – as P. vivax has very high prevalence but low morbidity and mortality. This was corrected through using ‘deaths’ instead of ‘cases’ as both are covered in the used refence.

51 – Malaria severity is really a spectrum and not distinct entities. If, however, it needs to be classified into groups, “simple malaria” is not a term that is used in the field. Simple malaria was removed from the group list.

53 - Age and immune status are also factors that are important in determining infection outcome and should be included in the list. Age and immune status were added to the list.

55 – The authors seem to be mixing the innate immune response with physical barriers. They are quite distinct. This has been corrected through separating immune response from physical barriers. A new reference was also added.

The malaria pathogenesis section is spotty; the input of a malaria expert as co-author is needed. SNPs and IL-21 are likely only minor players in pathogenesis and should not be mentioned in the absence of other factors (unless these are going to be tied into helminths or microbiota). Guided by our malaria expert added as new co-author, this has been corrected by removing the SNP/TLR table and adding inputs about key malaria pathogenesis factors from 2 reviews.

68-69 – “Environmental factors” seem to be conflated with genetics in this section. This was corrected by replacing the word ‘environmental’ with ‘genetic’.

88 – This statement about helminths “Being parasites” and therefore result in damage is backwards thinking. The characteristics of the organism is considered and then given a classification. This was corrected by removing that whole sentence.

In the paragraph starting Line 111 discussing helminths and malaria. There is a nice meta-review in PLoS NTD by Afolabi that should be discussed. Thank you, we have now used and cited this review.

173 – I do not see any reference for this statement. The referenced paper (43) does not look at the presence of this antibody and specifically states that the role of the anti-gal antibodies cannot be assessed in their study. We apologize for the confusion, the correct reference is the paper by B. Yilmaz et al. (now ref 48), “Article Gut Microbiota Elicits a Protective Immune Response against Malaria Transmission,” Cell, vol. 159, no. 6, pp. 1277–1289, 2014. this is now corrected and the authors stated in their highlights that " α-gal is expressed at the surface of Plasmodium sporozoites, Anti-α-gal Abs recognizing E. coli O86:B7 are protective against malaria, Anti-α-gal Abs are cytotoxic to Plasmodium sporozoites, Vaccination against α-gal confers sterile protection against malaria.

Lines 212 and following. This sentence is very awkwardly worded – I believe it is a double negative and therefore not stating what the authors wish. The sentence has been splitted into two to make statements clearer.

Line 221 to 223 – Were these fish parasitized by a helminth? This sentence and its relevance to the discussion is unclear. Corrected as follows: ‘In a study conducted on Gasterosteus aculeatus fish infected by Schistocephalus solidus – a natural parasitic helminth of fish, changes in the gut microbiota composition were shown to depend on parasite exposure’.

Table 2 – separate out by studies in mice first followed by all the studies in humans. Thank you for this suggestion, we have now modified the table.

Line 240 Studies have been conducted in vitro? I am not sure that you listed any of those. The majority of the studies seem to have been conducted in mice. This has been corrected.

Line 244 “gut” typo. Corrected.

Lines 245 - 247 Again the wording is awkward and not likely as meant. The sentence was paraphrased and made clearer.

Line 248 – I am not sure that this was the ‘most significant finding’. I believe the authors should describe all of the findings and allow the reader to make their own decisions as to what is significant. Corrected by detailing findings further for the reader to make their own decision.

Round 2

Reviewer 1 Report

I'm satisfied with changes

Reviewer 2 Report

The manuscript has been substantially improved. All of my concerns have been met.